# EFFICIENT TRANSFER LEARNING FROM ARBITRARY PRE-TRAINED MODELS

## ABSTRACT

Transfer learning typically involves loading pre-trained weights as an initialization, followed by fine-tuning on a downstream task. As pre-trained models become ever larger, this procedure is becoming prohibitively expensive, as we are forced to re-use the pre-trained architecture for fine-tuning. This procedure also precludes combining multiple pre-trained models that learn complementary information. Moreover, alternatives such as knowledge distillation do not reflect that we wish to transfer aspects of the pre-trained representation that are most relevant to the downstream task. To address these challenges, we introduce *Adaptive Feature Transfer* (AFT). Instead of transferring weights, AFT operates purely on features, thereby decoupling the choice of the pre-trained model from the possibly smaller downstream model. AFT (1) enables transfer from multiple pre-trained models, even over multiple modalities, with minimal training overhead and no inference overhead; (2) selectively transfers the information in the pre-trained features most relevant for the downstream task, through a prior that favors low mutual information between the downstream inputs and features given the pre-trained features; (3) performs feature transfer in an efficient kernel formulation that prioritizes the most relevant degrees of freedom. Empirically, AFT delivers a substantial boost in performance across diverse vision, language, and multi-modal datasets, relative to both standard transfer learning and knowledge distillation with the downstream model. Anonymous code for reproducing our results are available at `https://anonymous.4open.science/r/aft-6C30`.

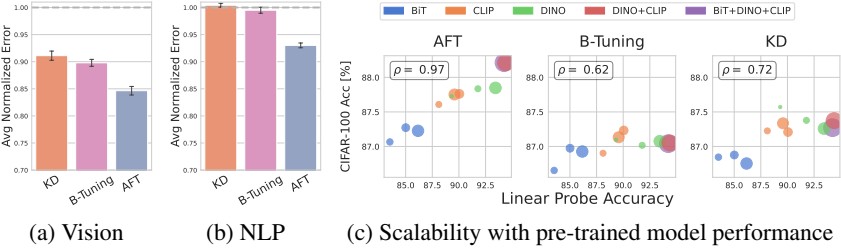

(a) Vision  (b) NLP  (c) Scalability with pre-trained model performance

Figure 1: Adaptive Feature Transfer (AFT) enables compute-efficient transfer learning from an arbitrary set of pre-trained models into a single downstream model, significantly outperforming competing methods including Knowledge Distillation (KD) and B-Tuning (You et al., 2022) when averaged over (**a**) 6 vision tasks and (**b**) 8 NLP tasks. (**c**) AFT performance correlates uniquely well with the quality of the pre-trained features, as measured by the linear probe accuracy. The marker size indicates pre-trained model size, ranging from 87M to 2.7B.

## 1 INTRODUCTION

Despite its increasing importance, transfer learning methodology has not kept up with the demands of modern deep learning. It remains the standard practice to simply start with a pre-trained parameter vector and then fine-tune on downstream data with the same architecture. As pre-trained models continue to grow in size (Bommasani et al., 2021; Brown et al., 2020; Dosovitskiy et al., 2020; Zhai et al., 2022), the computational burden of fine-tuning them drastically escalates to the point that many practitioners do not possess the resources to fine-tune state-of-the-art models in vision and language. Furthermore, this approach precludes transferring from multiple pre-trained models that

learn complementary information due to different pre-training strategies, when a variety of distinctly pre-trained models have become available in domains such as computer vision (Oquab et al., 2023; Radford et al., 2021; Kolesnikov et al., 2020; Chen et al., 2020) and language (Devlin et al., 2018; Sanh et al., 2020; Touvron et al., 2023).

To address these limitations, we propose *Adaptive Feature Transfer* (AFT), a highly efficient method to transfer from an arbitrary set of pre-trained models into a single downstream model within the compute budget of training only the downstream model. Based on the observation that the features from a well-pretrained models are likely to contain information highly relevant to downstream predictions, AFT introduces an informative prior favoring low mutual information between the downstream inputs and features given the pre-trained features. AFT then efficiently optimizes it by exploiting a kernel formulation of the objective. This approach empowers AFT to perform cross-architecture transfers and assimilate complementary information from multiple pre-trained models.

Across multiple vision, language, and multi-modal datasets, we show AFT delivers a substantial performance improvement compared to both standard transfer learning (STL) and alternatives such as Knowledge Distillation and B-Tuning (You et al., 2022). Moreover, we find AFT exhibits a high correlation between its performance and the quality of pre-trained features, measured by their linear probe accuracies, and a strong ability to harness complementary information learned by multiple pre-trained models (Figure 1).

## 2 RELATED WORK

**Transfer learning** Standard transfer learning proceeds by loading a pre-trained parameter vector as the initialization for parameters $\theta$ of a downstream model with the same architecture, followed by updating $\theta$ by minimizing the downstream loss $L(\theta)$, known as fine-tuning (Zhuang et al., 2019). This simple approach has enabled state-of-the-art performances on a wide range of vision (Dosovitskiy et al., 2020; Oquab et al., 2023; He et al., 2015) and language tasks (Devlin et al., 2018; Touvron et al., 2023). To extract additional useful information from the pre-trained model, Shwartz-Ziv et al. (2022) propose a Bayesian transfer learning approach. In addition to using the pre-trained initialization, this approach uses an approximate posterior for the pre-training data as an informative prior $p(\theta)$ for downstream learning, leading to improved performance across several vision datasets. Similar to standard transfer learning, this approach restricts the downstream model to have the same architecture as the pre-trained model, since it requires evaluating the approximate posterior of the pre-trained model at the downstream parameters $\theta$. Conceptually, the Bayesian transfer learning perspective points to a natural possibility of transferring across architectures or from many pre-trained models. This can be done by defining an informative prior that similarly facilitates the transfer of information learned by the pre-trained models without requiring the downstream model to have the same architecture.

**Knowledge distillation** Knowledge Distillation (KD) (Hinton et al., 2015) is a method that can be applied to compress a large model, referred to as the teacher model, to a smaller model, referred to as the student model, with the goal of minimizing performance degradation (Wang & Yoon, 2020). Traditionally, KD starts with a teacher $T$ trained on a dataset $\mathcal{D}$ and then trains the student $S$ to match the predictions of the teacher on the same dataset to achieve model compression. In the setting of transfer learning, this version of KD is generally not suitable for training a student to perform a novel downstream task, since the teacher does not predict the downstream targets (e.g. the classes may be different) and we therefore don't wish to match the student's prediction to the teacher's. Instead, we focus on the version of KD which trains the student to predict the teacher's features $\phi_T$, such as through a learned linear transformation $V$ applied to the student's feature $\phi_S$ under a regression objective $\mathbb{E}_{x\sim\mathcal{D}}\left[\|\phi_T(x) - V\phi_S(x)\|_2^2\right]$, where $V$ can account for the difference in dimensionality (Heo et al., 2019a; Huang & Wang, 2017; Heo et al., 2019b; Gu et al., 2023; Ahn et al., 2019). This procedure can be extended to use multiple teachers by simultaneously minimizing the sum of multiple KD objectives each with a different teacher, as proposed in Liu et al. (2020); Wu et al. (2021), equivalent to simultaneously predicting the concatenation of the teachers' features.

While KD is a natural candidate for *model compression*, its objective is fundamentally misaligned with the goal of *transfer learning*. Ahn et al. (2019) show that the feature space KD objective has an information-theoretic interpretation as minimizing $H(\phi_T|\phi_S)$ the conditional entropy of the teacher

features given the student features, which penalizes any information learned by the teacher but not by the student. Since the teacher was trained on a related but different pre-training task, we should only aim to transfer information useful for performing the downstream task, rather than compressing all information learned by the teacher into the student irrespective of its downstream relevance.

**Multi-Source Transfer Learning**  Lee et al. (2019) propose to learn a classifier defined as a weighted combination of frozen pre-trained features, where the weights are derived from non-linear maximal correlation analysis. Chang et al. (2022) uses a mixture of experts (MoE) model to combine complementary information across different models and datasets to address the issue of data scarcity in material sciences. These methods do not reduce the inference cost with large pre-trained models. Gu et al. (2023) proposes to transfer features from the teachers to the students layer by layer, allowing for multiple teachers and different architectures .You et al. (2022) proposes Bayesian Tuning (B-Tuning) to efficiently transfer from heterogeneous pre-trained models by encouraging the fine-tuned model to predict the approximate posterior predictive mean of a linear model with pre-trained feature extractors, a low dimensional projection of the pre-trained features. In addition, several works propose to rank and select pre-trained models or features for transferring to a specific downstream task (You et al., 2022; Fumero et al., 2023; Deshpande et al., 2021). These methods are complementary to and can be used together with our method, which aims to maximize transfer performance once a set of pre-trained models is chosen.

## 3 OUR METHOD: ADAPTIVE FEATURE TRANSFER

We now introduce Adaptive Feature Transfer (AFT), a method that enables transfer learning from a set of pre-trained models of arbitrary sizes and architectures into a single downstream model, with negligible compute overhead compared to only training the downstream model.

### 3.1 CONSTRUCTING AN INFORMATIVE PRIOR FROM PRE-TRAINED FEATURES

The core idea of AFT is to impose an informative prior on the downstream learning to favor making predictions based on information already present in the pre-trained features, as they are highly likely to contain useful knowledge for the downstream task. Specifically, let $\theta \in \mathbb{R}^P$ be the downstream model parameters, a random variable $X \in \mathbb{R}^{d_{\text{in}}}$ be the downstream inputs, $\Phi = \phi_\theta(X) \in \mathbb{R}^{d_\phi}$ be the features of the downstream model, $Y = W\Phi \in \mathbb{R}^{d_{\text{out}}}$ be the downstream model outputs, and $\Psi = \psi(X) \in \mathbb{R}^{d_\psi}$ be some fixed pre-trained features, formed by concatenating the last layer features from an arbitrary number of pre-trained. We encode our preference with a prior that favors low mutual information between downstream features $\Phi$ and the input $X$ conditioned on $\Psi$,

$$p(\theta) \propto \exp(-\beta I(\Phi; X|\Psi)), \tag{1}$$

where the $I(\Phi; X|\Psi)$ measures information about the input used by the model to generate downstream features $\Phi$ that is not present in the pre-trained features $\Psi$ and $\beta > 0$ controls the strength of this prior. The mutual information is given by

$$I(\Phi; X|\Psi) = H(\Phi|\Psi) - H(\Phi|X, \Psi) = \mathbb{E}_{\Phi,\Psi}[-\log p(\Phi|\Psi)] + c \leq \mathbb{E}_{\Phi,\Psi}[-\log q_\rho(\Phi|\Psi)] + c, \tag{2}$$

where $H(\Phi|X, \Psi)$ is some constant $c$ since $\Phi$ is deterministic given $X$ and we used a a variational distribution $q_\rho(\Phi|\Psi)$ with variational parameters $\rho$ to approximate the inaccessible conditional density $p(\Phi|\Psi)$ and bound the mutual information.

We then perform Maximum A Posteriori (MAP) estimation, which minimizes the resulting bound on the negative log posterior, equal to $L(\theta) + \beta R(\theta)$, where $L(\theta)$ is the unregularized loss (e.g. cross-entropy loss) and $R(\theta)$ is the bound on the mutual information given by

$$R(\theta) = \min_\rho \mathbb{E}_{\Phi,\Psi}[-\log q_\rho(\Phi|\Psi)], \tag{3}$$

where the expectation can only be estimated using training samples. The effect of optimizing this objective is to maximize the downstream data fit while minimizing the information in downstream features $\Phi$ that cannot be decoded from the pre-trained features $\Psi$ via the map $q_\rho(\Phi|\Psi)$,

after optimizing for variational parameters $\rho$. We consider a simple Gaussian parameterization $q_\rho(\Phi|\Psi) = \mathcal{N}(\Phi|\rho\Psi, I)$, where $\rho : \mathbb{R}^{d_\psi} \to \mathbb{R}^{d_\phi}$ is an affine transformation, which leads to:

$$R(\theta) = \min_\rho \mathbb{E}_{\Phi,\Psi}\left[\|\Phi - \rho\Psi\|^2\right], \tag{4}$$

after ignoring some $\theta$−independent constants. Since the minimization over the offsets in the affine transformation is equivalent to subtracting the mean from both $\Phi$ and $\Psi$, we will henceforth assume that $\Phi$ and $\Psi$ have been pre-processed to have zero-mean and assume $\rho \in \mathbb{R}^{d_\phi \times d_\psi}$ to be a linear transformation. Contrasting this objective with the KD objective, expressed in the current notations:

$$R_{\text{KD}}(\theta) = \min_V \mathbb{E}_{\Phi,\Psi}\left[\|V\Phi - \Psi\|^2\right], \tag{5}$$

with $V \in \mathbb{R}^{d_\psi \times d_\phi}$, we see that minimizing the KD objective requires the downstream $\Phi$ features to contain all information needed to predict the pre-trained features $\Psi$, while our objective $R(\theta)$ only requires the downstream features $\Phi$ to lie in the span of the pre-trained features $\Psi$, allowing for discarding information in $\Psi$. Therefore, when optimized together with the training loss, our objective $R(\theta)$ makes it much easier for the downstream model to selectively transfer only the task-relevant features from pre-training.

## 3.2 Improving the objective using the kernel

Estimating the regularization term $R(\theta)$ requires handling both optimization and statistical challenges: 1) since evaluating $R(\theta)$ requires finding the optimal variational parameters $\rho$, which changes every time we update $\theta$, we want to maximally simplify the optimization problem for $\rho$, and 2) since we wish to estimate the true $R(\theta)$, or equivalently the true $I(\Phi, X|\Psi)$, whose exact value is given by an expectation over the true rather than empirical distribution of $\Phi$ and $\Psi$, we want to avoid over-fitting to the training data when optimizing for $\rho$ when we replace the expectation in Eq. 4 with its empirical estimate.

In addition to the simplifying assumption on the form of $q_\rho(\Phi|\Psi)$, we now show how to exploit a kernel formulation of the objective to further mitigate both challenges. Recall that the behavior of a linear model $f(\cdot) = w^\top \phi(\cdot)$ is completely characterized by its kernel $k_\Phi(x, x') = \phi(x)^\top \phi(x')$. From a kernel perspective, the existence of $\rho \in \mathbb{R}^{d_\phi \times d_\psi}$ such that $\Phi = \rho\Psi$ is exactly equivalent to the existence of $\tilde{\rho} \in \mathbb{R}^{d_\phi \times d_\psi}$ such that $k_\Phi = k_{\tilde{\rho}\Psi}$. Therefore, in AFT we replace the $\ell_2$ distance between the features with a distance between their kernel functions

$$R_{\text{AFT}}(\theta) = \min_\rho \sqrt{\mathbb{E}\left[(k_\Phi(X, X') - k_{\rho\Psi}(X, X'))^2\right]}, \tag{6}$$

where $X$ and $X'$ are drawn from the input distribution. As with the previous objective in Eq. 4, this objective achieves a minimum value of 0 if and only if each $\phi_i(\cdot), i = 1, ..., d_\phi$, are in the span of $\{\psi_i(\cdot)\}_{i=1}^{d_\psi}$. However, the kernel formulation has the key advantage that part of the optimization problem over $\rho$ is done automatically since the kernel is invariant under any orthogonal transformation of the features, implying that we only need to optimize $\rho$ up to an orthogonal transformation, significantly reducing the complexity of the inner optimization.

To prevent over-fitting the variational parameters $\rho$ to the empirical distribution of the features, we parameterize $\rho$ as a diagonal matrix $\text{diag}(\sigma(s))$, i.e. $\rho_{ii} = \sigma(s_i)$, where $\sigma$ is the sigmoid function and $s$ is a $d_\psi$-dimensional vector. Note the ability to use a diagonal $\rho$ is a distinct advantage of the kernel formulation, which does not require the features to have the same dimensions. Using this parameterization, we greatly reduce the number of variational parameters to optimize, while retaining the ability for the model to weigh each dimension of the pre-trained features according to their task-relevance. Furthermore, thanks to using the kernel formulation, we are effectively searching over all $\rho' = U\rho = U\text{diag}(s)$, where $U$ is any orthogonal matrix, that map between pre-trained and downstream features, without actually optimizing the dense matrix $U$. Finally, we normalize the features to have unit $\ell_2$ norm before computing the respective kernels, i.e., $k_\Phi(x, x') := \phi(x)^\top \phi(x')/\|\phi(x)\|\|\phi(x')\|$, to reduce the variance in the entries of the kernel. In Section 4.5, we compare AFT with its other variants and show that both using the kernel formulation and learning a diagonal $\rho$ indeed improves its performance.

**Stochastic kernel distance estimation** For a practical implementation, we estimate $\delta(\theta, \rho) := \sqrt{\mathbb{E}\left[(k_\Phi(X, X') - k_{\rho\Psi}(X, X'))^2\right]}$ with a mini-batch estimate $\hat{\delta}(\theta, \rho) := \sqrt{\frac{1}{B^2}\sum_{i=1}^{B}\sum_{j=1}^{B}(k_\Phi(x_i, x_j) - k_{\rho\Phi}(x_i, x_j))^2} = \frac{1}{B}\left\|K^\Phi_{\text{batch}} - K^{\rho\Psi}_{\text{batch}}\right\|_F$, where $K^\Phi_{\text{batch}}$ and $K^{\rho\Psi}_{\text{batch}}$ are kernel matrices evaluated on a batch of $B$ inputs. We then perform gradient-based optimization jointly over $(\theta, \rho)$. Algorithm 1 details the training procedure using the SGD optimizer for simplicity. Note we compute and cache the pre-trained features on the training set once and simply retrieve them during training without spending additional time to compute them.

---

**Algorithm 1** Adaptive Feature Transfer (AFT)

---

**Require:** Pre-computed pre-trained features, downstream data, downstream model $f_\theta = W \circ \phi_\theta$, downstream loss function $L$, batch size $B$, learning rates $(\eta_1, \eta_2)$, regularization coefficient $\beta$

1: **for** each mini-batch $(X_{\text{batch}} \in \mathbb{R}^{B \times d_{\text{in}}}, Y_{\text{batch}} \in \mathbb{R}^{B \times d_{\text{out}}}, \Psi_{\text{batch}} \in \mathbb{R}^{B \times d_\psi})$ **do**
2: $\quad$ Compute features $\Phi_{\text{batch}} = \phi_\theta(X_{\text{batch}}) \in \mathbb{R}^{B \times d_\phi}$ and outputs $\hat{Y}_{\text{batch}} = \Phi_{\text{batch}}W^\top$
3: $\quad$ Scale pre-trained features $\Psi_{\text{batch}} \leftarrow \Psi_{\text{batch}}\rho^\top$
4: $\quad$ Subtract the mini-batch mean from $\Phi_{\text{batch}}$ and $\Psi_{\text{batch}}$ and normalize each row
5: $\quad$ Compute $B \times B$ mini-batch kernels $K^\Phi_{\text{batch}} = \Phi_{\text{batch}}\Phi^\top_{\text{batch}}, K^{\rho\Psi}_{\text{batch}} = \Psi_{\text{batch}}\Psi^\top_{\text{batch}}$
6: $\quad$ Compute mini-batch loss $\hat{L}(\theta) = L(\theta, Y_{\text{batch}}, \hat{Y}_{\text{batch}})$ and the kernel distance estimate:

$$\hat{\delta}(\theta, \rho) = \frac{1}{B}\left\|K^\Phi_{\text{batch}} - K^{\rho\Psi}_{\text{batch}}\right\|_F$$

7: $\quad$ Update $\theta$ and $\rho$ using SGD:

$$\theta \leftarrow \theta - \eta_1\nabla_\theta\left(\hat{L}(\theta) + \beta\hat{\delta}(\theta, \rho)\right), \quad \rho \leftarrow \rho - \eta_2\nabla_\rho\hat{\delta}(\theta, \rho)$$

8: **end for**

---

## 4 EXPERIMENTS

We evaluate our proposed method Adaptive Feature Transfer (AFT) across a variety of vision, language, and multi-modal datasets and compare with standard transfer learning (STL), Knowledge distillation (KD), and B-Tuning (You et al., 2022). All four methods start with the same pre-trained initialization of the downstream model, except that AFT, KD, and B-Tuning additionally optimize their respective regularization terms that enable transfer from one or multiple additional pre-trained models. A hyperparameter $\beta > 0$ is tuned on validation performance to optimally weigh the regularization term for each method. We include full experiment details, such as hyperparameter tuning in the Appendix A. We report the mean and standard errors computed across 3 runs for each method.

### 4.1 IMAGE CLASSIFICATION

**Effective transfer from SOTA vision foundation models** We evaluate AFT's ability to transfer from state-of-the-art vision foundation models into commonly used downstream architectures, including ViT-S (Dosovitskiy et al., 2020), MLP-Mixer-B (Tolstikhin et al., 2021), and ResNet-50 (He et al., 2015). We initialize the downstream models with ImageNet-1K checkpoints for all methods. In Figure 2a and 2b, we show performance when transferring from ViT-G DINOv2, the largest model in the DINOv2 family with over a billion parameters, on CIFAR-10 (Krizhevsky et al., 2009), CIFAR-100 (Krizhevsky et al., 2009), Oxford Flowers-102 (Nilsback & Zisserman, 2008), Oxford-IIIT Pets (Parkhi et al., 2012), Describable Textures Dataset (DTD) (Cimpoi et al., 2014) and Food-101 (Bossard et al., 2014) datasets. We find AFT significantly boosts the performance of all three models, reducing the error by an average of over 15% relative to STL performance (Figure 2a), and considerably outperforms KD and B-Tuning in most cases as well as on average.

**Transfer from multiple pre-trained models** In Figure 2c, we show the performance on CIFAR-100 when transferring from various vision foundation models, including BiT ResNet-101x3 (Kolesnikov et al., 2020) (denoted BiT), CLIP ViT-G (Radford et al., 2021) (denoted CLIP) and ViT-G DINOv2 (Oquab et al., 2023) (denoted DINO). AFT yields large improvements over STL and significantly outperforms all other competing methods except for ResNet-50, where KD is better

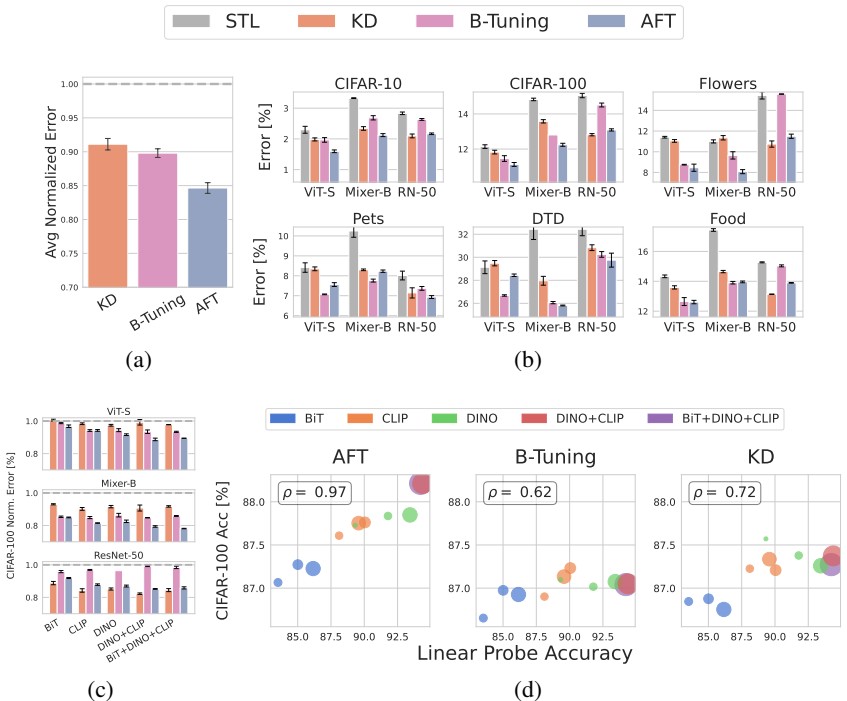

Figure 2: Evaluation on 6 vision datasets using ViT-S, MLP-Mixer-B, and ResNet-50 as downstream models. (**a**) AFT achieves a significantly lower normalized error, averaged across 6 datasets and 3 downstream models when transferring from ViT-G DINOv2. The error is normalized by the STL error before averaging. (**b**) Breakdown of unnormalized error for each downstream model and dataset. (**c**) Effect of transfer from different pre-trained models and their combinations on CIFAR-100. AFT achieves the best performance when combining features from multiple pre-trained models (DINO + CLIP or BIT + DINO + CIP). (d) Downstream accuracy versus linear probe accuracy of pre-trained features for AFT, B-Tuning, and KD, averaged across 3 downstream models on CIFAR-100. AFT yields consistent performance gains as we improve the quality of the pre-trained features, showing the highest correlation with the linear probe accuracy. The marker size represents the number of parameters in the pre-trained models, ranging from 87 million to 2.7 billion.

by a small margin compared to AFT. AFT consistently achieves the best performance by transfer from multiple pre-trained models such as DINOv2 + CLIP or BIT + DINOv2 + CLIP, suggesting that AFT is leveraging complementary features learned by these models due to different inductive biases, pre-training objectives, and pre-training data. For example, while CLIP is trained with a contrastive objective for matching images to texts, DINOv2 is trained with pure self-supervision without text information, and BiT is fully supervised and uses a ResNet architecture rather than a ViT. Consequently, each model is likely to learn useful but different visual features that contain complementary information relevant to the downstream task. On the other hand, combining pre-trained features from multiple models can lead to rapid growth in the amount of redundant or irrelevant features, necessitating an adaptive approach that can identify and only transfer the most relevant subset for the task. In Section 4.4, we show AFT indeed adaptively reweights the features depending on the pre-trained models provided. By contrast, in Figure 2c, we find that KD, which aims to distill all information learned by the pre-trained models, is unable to benefit from using multiple of them.

**Predictable performance scaling** As AFT biases the final-layer linear predictor to use task-relevant features from the pre-trained models, we expect its performance to correlate with the quality of pre-trained features, as measured by their linear probe accuracy (accuracy of a linear classifier using those features). Indeed, Figure 2d shows a strong correlation between the two, demonstrating that 1) AFT is effective at transferring the kernel formed by the features of the pre-trained kernel, and 2) AFT will achieve better performance with pre-trained models that learn more useful features for the downstream task. As a result, we can predict for which pre-trained model(s) AFT will likely

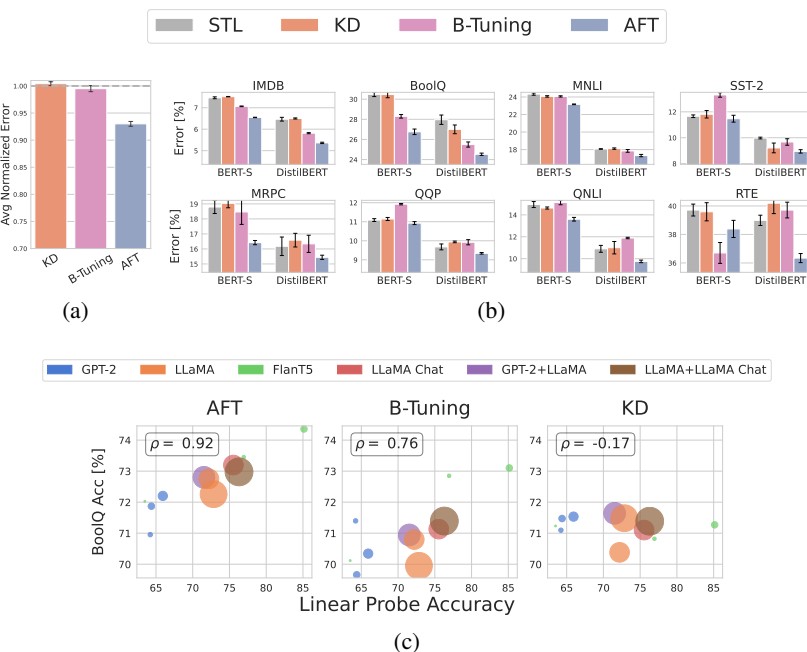

(a)                    (b)

(c)

Figure 3: Evaluation on 8 language dataset using BERT Small and DistillBert as downstream models. (a) AFT achieves a significantly lower normalized error, averaged across 6 datasets and 2 downstream models when transferring from Flan-T5 Large. The error is normalized by the STL error before averaging. (b) Breakdown of unnormalized error for each downstream model and dataset. (c) Downstream accuracy versus linear probe accuracy of pre-trained features for AFT, B-Tuning, and KD, averaged across both downstream models on BoolQ. AFT yields consistent performance gains as we improve the quality of the pre-trained features, showing the highest correlation with the linear probe accuracy. The marker size is proportional to the number of parameters in the pre-trained models, ranging from 61M to 14B.

achieve the best performance, by evaluating their linear probe accuracies, greatly simplifying the selection of the pre-trained model(s) in practice. Indeed, we could have correctly predicted in every setting that transferring from ViT DINOv2 + ViT CLIP would outperform transferring from either by noting that the combination of both models has a higher linear probe accuracy than either model. By comparison, other methods' performance is less well correlated with the linear probe accuracy, which explains why they don't benefit from transferring multiple models and provides strong evidence to our claim that AFT is a superior approach to transfer learning that should scale better as we use larger and better pre-trained models. While the linear probe accuracy of a sufficiently large pre-trained model can exceed the accuracy of AFT, the former is only efficient to train (via logistic regression) but still expensive to deploy, as it requires inference with the original pre-trained model, and is therefore not a viable alternative to the methods considered here. For example, the linear probe accuracy of ViT-L CLIP roughly matches AFT accuracy when transferred to ViT-S on CIFAR-100, but ViT-L CLIP has 428M parameters, 20 times larger than ViT-S.

## 4.2 NATURAL LANGUAGE PROCESSING

We explore transferring from some of the strongest open-source large language models, including GPT-2 (Radford et al., 2019), Flan-T5 (Chung et al., 2022), and LLaMA 2 (Touvron et al., 2023), into much smaller ones: BERT Small (Devlin et al., 2018) and DistillBERT (Sanh et al., 2020). In language models, there is no exact analog of last-layer features at the input level since the model maintains an embedding for each token. As such, we follow the common practices for extracting input (i.e. sequence) level features for the following models used in our evaluation as follows: we use the embedding of the [CLS] token for BERT models, and the decoder's embedding of the last token for GPT-3, Flan-T5, and LLaMA.

In Figure 3a and 3b, we show the performance of AFT and competing methods at transferring from Flan-T5 Large to BERT Small and DistillBERT on the following 8 datasets: Large Movie Review (IMDB) (Maas et al., 2011), BoolQ (Wang et al., 2019), MNLI (Williams et al., 2018), SST-2 (Socher et al., 2013), MRPC (Dolan & Brockett, 2005), QQP (Wang et al., 2018), QNLI (Rajpurkar et al., 2016) and RTE Wang et al. (2018). AFT significantly outperforms the competing methods.

Similar to the case for vision, we find AFT's performance scales with a strong correlation with the linear probe accuracy of pre-trained features, as shown in Figure 3c, whereas other methods have a much lower correlation. In addition, we find using AFT with pre-trained language models with instruction-tuning, like Flan-T5 and LLaMA Chat, led to the best performance after transfer, in line with their superior zero-shot question answering capabilities (Chung et al., 2022).

Unlike in vision datasets, we find combining multiple pre-trained models often leads to no improvement in AFT's performance, as shown in Figure 3c. However, this behavior is not surprising since combining these pre-trained models does not increase the linear probe accuracy either, suggesting there is little complementary and non-overlapping information learned between these pre-trained language models. A natural explanation here is that these pre-trained large language models are all highly similar to each other in the pre-training datasets, objectives, and architectures, since they are all transformer-based generative models trained predominantly with next or masked token prediction on a similar distribution of text from the internet.

### 4.3 MULTI-MODALITY

The capability to efficiently transfer from multiple models naturally positions AFT for use in multi-modal applications. In these settings, the architecture typically includes modality-specific sub-components, like an image encoder and a text encoder. Since pre-trained models with strong performance often exist for each individual modality, we expect AFT can boost multi-modal performance by transferring the complementary, modality-specific features learned by these models. To illustrate this possibility, we consider SNLI-VE (Xie et al., 2019; 2018), a challenging visual entailment dataset where the objective is to determine if a given text accurately corresponds to an image, with the possible classes being positive, negative, or neutral. We use the smallest version of CLIP as the downstream model, which consists of a ResNet-50 image encoder and a transformer text encoder, initialized to the trained checkpoint. From the image features $\phi_I(x_I)$ and text features $\phi_T(x_T)$, we construct a classifier $f_\theta(x_I, x_T) = W\phi(x_I, x_T)$ whose features $\phi(x_I, x_T)$ is given by the (flattened) tensor product $\phi_I(x_I) \otimes \phi_T(x_T)$, which represent the pairwise interactions between the image and text features and enable computations such as $\phi_I(x_I)^\top \phi_T(x_T)$, a measure of semantic similarity between the image and text due to the CLIP pre-training. In Table 1, we find that AFT can improve CLIP's performance on this task by simultaneously transferring from a ViT-L trained with DINOv2 and LLaMA 13B and again outperforms KD.

Table 1: AFT improves CLIP's accuracy on SNLI-VE by transfer from DINOv2 and LLaMA 13B.

| Method | STL | KD | AFT |
|---|---|---|---|
| SNLI-VE Acc. | $73.69_{\pm0.28}$ | $74.05_{\pm0.05}$ | $\mathbf{74.39_{\pm0.18}}$ |

### 4.4 VISUALIZING LEARNED FEATURE WEIGHTING IN $\rho$

In Figure 4a, we show the distribution of learned feature weights $\rho_i$ at convergence on CIFAR-100 with ViT-S as the downstream model and pre-trained models from the set {BiT, DINO, CLIP}. AFT indeed learns non-uniform weighting for individual features ($\rho_i$ is initialized to 0.5 for all $i$). When transferring from all three models, AFT learns to upweight CLIP and DINO features and downweight BiT features, in line with our finding in Figure 2c that adding BiT to DINO and CLIP features did not improve further transfer performance.

In Figure 4b, we show the weights learned when we transfer from DINO and a random noise model whose features contain no useful information and are sampled from $\mathcal{N}(0, I_{d_{\text{noise}}})$, where $d_{\text{noise}} = 2048$ is the feature dimension of the noise model. AFT successfully assigns much smaller weights to the noise features so that the performance is unaffected by their presence, as shown in Figure 4c. By contrast, KD performance quickly degrades to near STL level as we introduce the noise features.

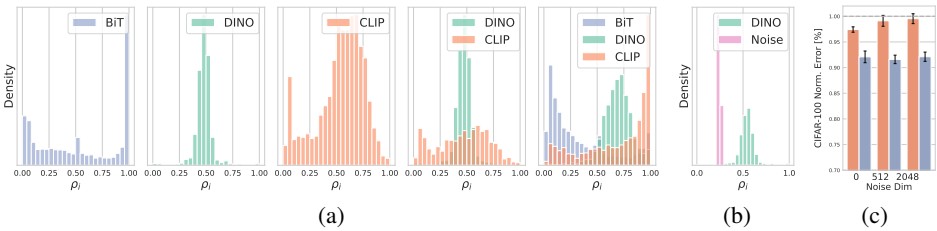

Figure 4: (a) Distribution of learned feature weights $\rho$ for each pre-trained model. The legend shows which pre-trained models are simultaneously used. (b) Distribution of $\rho$ in the presence of random noise features. (c) AFT performance as a function of noise dimensions.

## 4.5 ABLATION EXPERIMENTS

We investigate the impact of key design choices in AFT on its performance on CIFAR-100 and BoolQ dataset. We compare AFT with four other variants where a) we do not use a kernel formulation and directly use the objective listed in Eq. 4 as a regularization, b) the ability to learn a diagonal $\rho$ is disabled, causing it to default to identity, c) we replace the linear kernel $k(x, x') = \phi(x)^\top \phi(x')$ with radial basis function (RBF) kernel $k(x, x') = \exp\left(-\|\phi(x) - \phi(x')\|^2\right)$, or d) we perform bi-level optimization over $\theta$ and $\rho$ by performing 5 inner updates for $\rho$ per update of $\theta$. We find using the kernel formulation and learning the feature weights $\rho$ are essential to AFT's performance, while the use of alternative kernels such as the RBF kernel and bi-level optimization does not impact the performance in any significant way.

We also investigate the effectiveness of AFT in data-scarce scenarios by sub-sampling the CIFAR-100 and BoolQ training set. AFT remains the most effective method cross training set sizes.

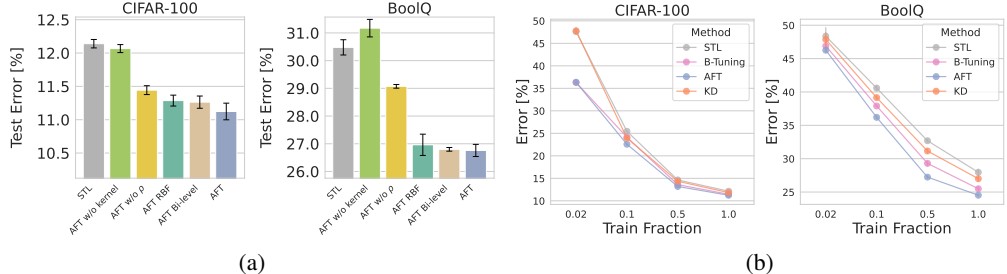

Figure 5: (a) Ablation studies: using the kernel and learning $\rho$ are the most essential contributors to AFT's performance. (b) AFT is the best performing method across data set sizes.

## 5 CONCLUSION

Our work addresses an important and timely problem in transfer learning: how to efficiently transfer from the variety of pre-trained models, each requiring increasingly large compute budgets to directly fine-tune and perform inference with, into a single smaller downstream model. To do so, we propose AFT, a novel method for transfer learning that accurately reflects the reality that not all the pre-trained features will be relevant to the downstream task. As a result, AFT is fundamentally more well-suited for transfer learning than Knowledge Distillation, which transfers information irrespective of its relevance to the downstream task. Through an extensive evaluation with various state-of-the-art pre-trained models and downstream models on 15 datasets across vision, language, and vision-language tasks, we show AFT significantly outperforms the competing methods across the board and benefits considerably more from stronger pre-trained models.

We hope our work enables the community to more effectively leverage large pre-trained models that have otherwise been prohibitively expensive to use.

## REPRODUCIBILITY STATEMENT

We provide a self-contained anonymous code base for reproducing all results at `https://anonymous.4open.science/r/aft-6C30`. We also provide training details including the hyperparameter grid, optimizer, and data preprocessing in Appendix A. We have carefully checked that the method description presented in Section 3 correctly corresponds to our implementation.

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

# A  TRAINING DETAILS

In all experiments, we tune the hyperparameter $\beta$ for AFT, KD, and B-Tuning by holding out 10% of the original training set and choosing the $\beta$ that achieves the highest holdout accuracy. We subsequently train with that $\beta$ on all of the training set and report test performance.

## A.1  VISION EXPERIMENTS

We use the timm (Wightman, 2019) implementation for all vision models, their pre-trained checkpoints, and data preprocessing pipelines. We do not use data augmentation in any experiment.

We use the Adam optimizer in all experiments and train for 50,000 steps with a batch size of 128 and a cosine lr decay schedule. We use a base learning rate of $1e-4$ for ViT-S and MLP Mixer-B, and $1e-3$ for ResNet-50. We tune $\beta \in \{3, 10, 30\}$ for AFT, $\beta \in \{0.1, 1, 10\}$ for KD, and $\beta \in \{1, 1e2, 1e3, 1e4\}$ for B-Tuning. We use the Adam optimizer and a learning rate of $1e-2$ for updating the vector $s$ parameterizing the diagonal elements of $\rho$.

## A.2  LANGUAGE EXPERIMENTS

We use the Hugging Face implementation of all the language models. We use the Adam optimizer in all experiments and train for 50,000 steps with a batch size of 64 and a cosine lr decay schedule. We use a base learning rate of $2e-5$ for both BERT Small and DistilBERT. We tune $\beta \in \{1, 3, 10\}$ for AFT, $\beta \in \{0.01, 0.1, 1\}$ for KD, and $\beta \in \{1, 1e2, 1e3, 1e4\}$ for B-Tuning. We use the Adam optimizer and a learning rate of $1e-2$ for updating the vector $s$ parameterizing the diagonal elements of $\rho$.

## A.3  SNLI-VE EXPERIMENTS

We use the Hugging Face implementation of CLIP ResNet-50. We use the Adam optimizer in all experiments and train for 1 epoch with a batch size of 64. We use a base learning rate of $1e-5$ for CLIP ResNet-50. We tune $\beta \in \{1, 3, 10\}$ for AFT, and $\beta \in \{0.01, 0.1, 1\}$ for KD. We use the Adam optimizer and a learning rate of $1e-2$ for updating the vector $s$ parameterizing the diagonal elements of $\rho$.

## A.4  EXTENDED RESULTS

Table 2: Unnormalized results for transfer to ViT-S in Figure 2c.

| Method | BiT | CLIP | DINO | DINO+CLIP | BiT+DINO+CLIP |
|---|---|---|---|---|---|
| KD | $87.79_{\pm 0.07}$ | $88.06_{\pm 0.06}$ | $88.17_{\pm 0.06}$ | $87.96_{\pm 0.21}$ | $88.13_{\pm 0.01}$ |
| B-Tuning | $88.01_{\pm 0.05}$ | $88.57_{\pm 0.06}$ | $88.54_{\pm 0.11}$ | $88.66_{\pm 0.13}$ | $88.67_{\pm 0.04}$ |
| AFT | $88.25_{\pm 0.09}$ | $88.56_{\pm 0.06}$ | $88.88_{\pm 0.06}$ | $89.23_{\pm 0.10}$ | $89.14_{\pm 0.00}$ |

Table 3: Unnormalized results for transfer to MLP-Mixer in Figure 2c.

| Method | BiT | CLIP | DINO | DINO+CLIP | BiT+DINO+CLIP |
|---|---|---|---|---|---|
| KD | $86.21_{\pm 0.05}$ | $86.63_{\pm 0.13}$ | $86.42_{\pm 0.11}$ | $86.55_{\pm 0.27}$ | $86.40_{\pm 0.06}$ |
| B-Tuning | $87.34_{\pm 0.06}$ | $87.42_{\pm 0.10}$ | $87.20_{\pm 0.16}$ | $87.43_{\pm 0.02}$ | $87.27_{\pm 0.04}$ |
| AFT | $87.40_{\pm 0.03}$ | $87.92_{\pm 0.02}$ | $87.76_{\pm 0.11}$ | $88.23_{\pm 0.07}$ | $88.42_{\pm 0.02}$ |

Table 4: Unnormalized results for transfer to ResNet-50 in Figure 2c.

| Method | BiT | CLIP | DINO | DINO+CLIP | BiT+DINO+CLIP |
|---|---|---|---|---|---|
| KD | $86.64_{\pm 0.15}$ | $87.32_{\pm 0.16}$ | $87.18_{\pm 0.10}$ | $87.62_{\pm 0.07}$ | $87.29_{\pm 0.14}$ |
| B-Tuning | $85.57_{\pm 0.10}$ | $85.42_{\pm 0.04}$ | $85.49_{\pm \text{NaN}}$ | $85.06_{\pm 0.05}$ | $85.19_{\pm 0.11}$ |
| AFT | $86.17_{\pm 0.05}$ | $86.78_{\pm 0.07}$ | $86.91_{\pm 0.09}$ | $87.18_{\pm 0.04}$ | $87.08_{\pm 0.10}$ |

