# OpenReview forum: "Efficient Transfer Learning from Arbitrary Pre-Trained Models"
_ICLR.cc/2024/Conference — Submitted to ICLR 2024_

### Official Review · Reviewer_ocxo · 2023-10-29

**Soundness:** 3 good
**Presentation:** 3 good
**Contribution:** 3 good
**Rating:** 6
**Confidence:** 4

**Summary:**

The motivation that inspired the work is important:  the pre-trained models are highly complex and difficult to fine-tune. However when you must train a model on a downstream task it could be that all the features that were learned on the source task are not necessary. So, if one had a way to select which features are relevant, then one could reduce the number of features needed to solve the downstream task and thus deploy smaller models. To address this task, they propose to  impose an L2 regularization which forces to find the most relevant features for the downstream task among all the features of the pre-trained models.

**Strengths:**

The idea of using the features learned in different models on the same data points and merge them together to represent the input is nice and, to the best of my knowledge, novel. It also makes sense performing an automatic feature selection in that space, in order to select the feature combination which is more informative. The experimental result presented in support of the idea are partially convincing.

**Weaknesses:**

I understand the attempt of providing a justification of the regularization loss using information theoretical bonds, but the way the authors arrive to the final form of the regularization they use, which is just a kernel version of the L2, eq. 9, is in my opinion unnecessarily involved and might create confusion. I suggest moving the text from eq 2 to eq 9 to the appendix.


MAJOR:  It is  not very clear why this form of R should help avoiding redundancy: the sigmoid function can set to zero irrelevant features, but if several features are simultaneously relevant (but correlated), I expect the solution  will not be sparse, but it will contain weights contributions from all. I suspect that this might lead to overfitting in data-scarce scenarios (see below).

Given the topic of the article I would have expected a comparison with LORA (https://arxiv.org/abs/2106.), where the features for the downstream task are selected by multiplying the original features by a low rank matrix before downstream fine tuning.

**Questions:**

Since the focus of the paper is transfer learning, which typically happens towards data-scarce tasks, I would have liked to see if the procedure is robust with respect to aggressive decimation of the target task. What happens if one attempts to use ~100 examples for category, as typical in clinical image analysis applications?

The last paragraph of page 4 is not very clear. The variational parameters are the components of the vector s, which, via the sigmoidal function, set the weight of the corresponding psi component in the rhoPsi kernel?

Minor: when they present the setting on page 3, the labels seem to be linear regression labels, while the experiments are on classification datasets. Please clarify.

---

> ### Author Response · Authors · 2023-11-21
> **Author Response**
>
> We appreciate your constructive feedback and recognition for the importance of our work. To address your questions, we run several new experiments to show that AFT indeed learns to downweight less useful features and is the best-performing method even under data scarcity. The new results are included in the updated version of the paper. We hope you can consider raising your score in light of our response.
>
> **On learning to downweight less useful features.**
> By allowing the model to freely weigh the pre-trained features via $\rho$ and ignore an arbitrary set of features without incurring any penalty, AFT incentivizes the model to prioritize learning the most useful pre-trained features, as they are most helpful for fitting the downstream data, and downweight the less useful ones. For this reweighting to be useful, the learned $\rho$ need not be sparse as long as it assigns sufficiently higher weights to the most relevant features.
>
> We provide two pieces of evidence that AFT indeed learns to downweight less useful features. First, we show that AFT learns to downweight the features from a pre-trained model that is less useful for the downstream task. In Figure 4(a), when we use BiT, DINO, and CLIP as the pre-trained models, AFT successfully learns to upweight CLIP and DINO features and downweight BiT features, which has considerably lower linear probe accuracy. Second, we run a new experiment where we append random noise features to DINO features and find that AFT learns to significantly downweights the noise features (Figure 4b), and thereby maintains its performance as we increase the number of noise features (Figure 4c), whereas KD performance quickly degrades.
>
> **On data scarcity.**
> Many of our datasets are already very small in size: Oxford flowers (1K), Oxford pets (3.6K), RTE (2.5K), MRPC (3.7K), and AFT leads to significant gains on these datasets (Figure 2(b) and Figure 3(b)). Furthermore, in Figure 5(b), we present an additional experiment where we subsample the training data down to 10 examples per class on CIFAR-100, and 100 examples per class on BoolQ. We show that AFT continues to outperform competing methods even in extreme data-scarce settings.
>
> **On LoRA.** LoRA is a method to efficiently fine-tune the original pre-trained model, but does not transfer to a smaller model and therefore still requires running expensive inferences with the original pre-trained model. As a result, LoRA is not a viable alternative to AFT or KD as a method to efficiently transfer from larger models into smaller ones. On the other hand, LoRA is entirely complementary to AFT as it can be used together during AFT fine-tuning to further reduce training cost.
>
> **On arriving at the final objective.** We included a detailed derivation of the objective starting with the bound on the mutual information to highlight the conceptual difference between AFT and KD. While the final objective is indeed a kernelized version of the L2 objective in Eq 9 (now Eq 4), it is fundamentally different from the similar-looking L2 objective in Eq 10 (now Eq 5) which KD optimizes. However, we appreciate the suggestion for a more concise presentation and have compressed the derivation.
>
> **On clarifying our notations**
> The $d_\psi \times d_\psi$ matrix $\rho$ is parameterized by the $d_\psi$ dimensional vector $s$ via $\rho = \mathrm{diag}(\sigma(s)),$ i.e. $\rho_{ii} = \sigma(s_i),$ where $\sigma$ is the sigmoid function. We have clarified this in the text.
>
> We did not assume that the labels $Y$ are linear regression labels. The equation $Y=W \Phi$ states that the output of the model is a linear transformation of its last layer features $\Phi,$ which holds for all neural networks and in both regression and classification tasks (where $Y$ would be the logits). Similarly, Algorithm 1 is applicable to both regression and classification tasks and we have updated the text to show that the loss $L$ is any loss function specified for the downstream task.

---

> > ### Comment · Reviewer_ocxo · 2023-11-23
> >
> > I am  happy of the explanation on LoRA and on the regression vs classification.  However, my question on what happens if several features are simultaneously relevant but correlated, was not answered. I therefore keep the score as it was.

---

> ### Author Response · Authors · 2023-11-23
> **Author Response**
>
> Thank you for responding! We’d like to clarify our answer to this question. If multiple features are simultaneously relevant but correlated, the weights learned by AFT indeed may not be sparse, nor have we claimed that they would be.
>
> However, assigning non-zero weights to correlated features should not harm AFT’s performance. In fact, we can show that if non-zero weights $\rho_a$ and $\rho_b$ are assigned to two duplicated features $\psi_a$ and $\psi_b$ for $a\neq b,$ the effect is exactly identical to having assigned a sparse $\rho'$ where $\rho'_a = \sqrt{\rho^2_a + \rho^2_b}$, $\rho'_b = 0$, and $\rho'_i = \rho_i$ for $i\notin\{a,b\}.$
>
> To prove this, it suffices to show that the target kernel $K_{\rho\Psi}(x,x') = (\rho \psi(x))^\top (\rho \psi(x'))$ in Eq 6 constructed by AFT with non-sparse weights $\rho$ is identical to the kernel constructed by AFT with sparse weights $\rho':$ $K_{\rho'\Psi}(x,x') = (\rho' \psi(x))^\top (\rho' \psi(x')).$
>
> This follows from direct calculation: $K_{\rho\Psi}(x,x') = (\rho \psi(x))^\top (\rho \psi(x')) = \sum_{i} \rho^2_i \psi_i(x)\psi_i(x') = \sum_{i\notin\{a,b\}} \rho^2_i \psi_i(x)\psi_i(x') + \rho^2_a \psi_a(x)\psi_a(x') + \rho^2_b \psi_b(x)\psi_b(x') = \sum_{i\notin\{a,b\}} \rho^2_i \psi_i(x)\psi_i(x') + (\rho^2_a + \rho^2_b) \psi_a(x)\psi_a(x') $
>
> By definition of $\rho'$, the above is equal to $\sum_{i\notin\{a,b\}} \rho^2_i \psi_i(x)\psi_i(x') + \rho'^2_a \psi_a(x)\psi_a(x') = (\rho' \psi(x))^\top (\rho' \psi(x')) = K_{\rho'\Psi}(x,x').$ The argument can be straightforwardly extended to an arbitrary number of duplicated features. Therefore, due to using the kernelized objective, assigning non-zero weights to duplicated features is automatically equivalent to assigning sparse weights in AFT.
>
> We hope you find this answer satisfactory and will consider revising your score!

---

> > ### Comment · Area_Chair_QUBP · 2023-12-02
> >
> > Dear Reviewer ocxo,
> >
> > The latest response from the authors has diligently sought to address your remaining question. Your feedback on whether it has been well solved would be greatly appreciated. Thanks for your time and support.
> >
> > Best,
> >
> > AC

---

### Official Review · Reviewer_U9CF · 2023-10-31

**Soundness:** 2 fair
**Presentation:** 2 fair
**Contribution:** 2 fair
**Rating:** 3
**Confidence:** 4

**Summary:**

This paper proposes Adaptive Feature Transfer (AFT) to transfer from an arbitrary set of pre-trained models into a single downstream model. When fine-tuning the downstream model, AFT introduces an informative prior favoring low mutual information between the downstream inputs and features given the pre-trained features. It then efficiently optimizes it by exploiting a kernel formulation of the objective. This paper conducts experiments on multiple vision, language, and multi-modal datasets, and AFT outperforms standard transfer learning and knowledge distillation methods.

**Strengths:**

1 This paper explores an interesting problem of efficient transfer learning from arbitrary pre-trained models.

2 The proposed AFT method is efficient and easy to implement. It is evaluated on multiple datasets on various tasks, including vision, language, and multi-modal, and outperforms standard fine-tuning and knowledge distillation methods.

3 This paper is clearly written and presented, and the proposed method is easy to follow.

**Weaknesses:**

1 Compared with the knowledge distillation mentioned in this paper (KD), the authors emphasize the contribution that KD transforms the downstream (student) features, while the proposed AFT transforms the pre-trained (teacher) features. However, in the general feature-based knowledge distillation framework [1], both teacher and student features can be transformed before minimizing their distances. This makes the proposed method a simple variant in the feature-based knowledge distillation framework and thus lack novelty.

2 Some related works are missing in this paper, including those improving standard transfer learning and those considering transfer learning from multiple pre-trained models. For example, [2] also proposes to match pre-trained features and downstream features during transfer learning. [3] and [4] also consider transfer learning from multiple pre-trained models and propose to use features or knowledge distillation from pre-trained models. More related works in these two topics should be discussed in the paper. In experiments, some of these more advanced transfer learning methods should be compared, instead of only comparing AFT with standard transfer learning or knowledge distillation.

3 Some issues in the experiments.

(1) It seems that in this paper, the pre-trained models are stronger than downstream models. Figures 2(c) and 3(c) also show that transfer learning by directly using pre-trained models leads to better results than AFT. This makes the problem setting in the experiments less convincing, especially considering that the linear probe from pre-trained models is also efficient.

(2) It is good to see experiments from vision, language, and multi-modal tasks, but in each task, only a few datasets are evaluated, and most of them seem to be easy.

(3) Transfer learning from multiple models is interesting, but currently, the number of models in the experiments is still small, and the improvements by using more pre-trained models are not clear from the results.

[1] Knowledge Distillation: A Survey. 2021

[2] Delta: Deep learning transfer using feature map with attention for convolutional networks. ICLR 2019

[3] Knowledge flow: Improve upon your teachers. ICLR 2019

[4] Ranking and Tuning Pre-trained Models: A New Paradigm of Exploiting Model Hubs. JMLR 2022

**Questions:**

1 What are the exact results before normalization in Figure 2(b)?

2 Could the kernel method in Section 3.2 still improve the performance if the downstream datasets have more training data? It would be better to have more experiments on more datasets or situations to validate the efficacy of such a design.

---

> ### Author Response · Authors · 2023-11-21
> **Author Response (Part 1)**
>
> Thank you for your thoughtful review of our work. Following your suggestion, we have added comparisons to a stronger baseline (B-Tuning in ref [4] you provided) and significantly expanded the scope of our evaluation to include 8 more datasets. The new results are included in the updated paper. We put a significant effort into these new experiments, inspired by your comments. We hope you can consider raising your score in light of our response. We respond to your questions below.
>
> **On comparison with related work.** Following your suggestion, we have added B-Tuning, the method proposed in [4], as a stronger baseline throughout our experiments. In Figure 2(a) and 3(a), we show that AFT significantly outperforms B-Tuning on average across all 6 vision datasets and 8 language datasets. Similar to KD, we find B-Tuning benefits considerably less than AFT from better pre-trained models, as shown in Figure 2(c) and Figure 3(c). We also added a discussion of other related works to the paper.
>
> **On expanding our evaluation.** Following your suggestion, we include 2 additional vision datasets and 6 additional language datasets, bringing the total number of datasets to 15, covering a comprehensive set of standard benchmarks for transfer learning. We report the averaged results in Figure 2(a) and Figure 3(a), showing that AFT significantly outperforms competing methods.
>
> **On using larger downstream datasets.** We've evaluated on 5 new downstream datasets that are larger than any dataset we previously used, including QQP (364K), MNLI (393K), QNLI (105K), Food101 (76K), and SST-2 (67.3K). AFT is the best performing method for all downstream models on 4 of these 5 datasets, verifying that AFT is effective even when the downstream datasets are large.
>
> **On novelty.** We would like to clarify that AFT has significant novelty relative to KD methods. AFT and KD have fundamentally different goals: KD aims to transfer all information in the pre-trained (teacher) features into the downstream (student) features, irrespective of whether it is relevant to the downstream task. By contrast, AFT aims to transfer only a subset of the information in the pre-trained features and allows the model to freely select which features to transfer, reflecting the fact that not all information in the pre-trained features is expected to be relevant to the downstream task. To illustrate the extent of this difference, consider an extreme example where the downstream features are identically zero and thus contain no information from the pre-trained features. The regularization term in Eq 6 which AFT optimizes is at its minimum and exactly zero, by setting $\rho = 0,$ since the downstream features are perfectly predictable from pre-trained features. By contrast, the regularization term in Eq 5 which KD optimizes will be at its maximum.
>
> The main novelty of AFT is in the specification of a novel prior for transfer learning that accurately reflects our belief that not all the pre-trained features will be relevant to the downstream task -- they are only a collection of features in which some task-relevant features are likely to be included. The specific act of transforming the pre-trained features instead of downstream features is a consequence of this distinct prior, and not the fundamental reason why AFT is novel.
>
> Furthermore, while Equation 4 in the reference you provided [1] presents a generic KD loss where both student and teacher features are allowed to be transformed, none of the specific methods mentioned there optimize for the selective transfer of knowledge by allowing the student to learn to freely ignore some of the teacher features in the manner that AFT does. Therefore, we believe that AFT is a novel method that is distinct from existing feature-based KD methods.
>
> **On AFR vs linear probe on pre-trained features.**
> Simply using a linear model with the pre-trained features is not a viable alternative, as it requires running expensive inference with the original pre-trained model, the very problem that AFT (and KD) aims to solve by transferring to much smaller models. For example, the linear probe accuracy of ViT-L CLIP roughly matches AFT accuracy when transferred to ViT-S on CIFAR-100. But ViT-L CLIP has 428M parameters, 20 times larger than ViT-S. We have added this discussion to the paper.

---

> > ### Author Response · Authors · 2023-11-21
> > **Author Response (Part 2)**
> >
> > **On whether more pre-trained models should always help.**
> > While AFT enables transfer from multiple pre-trained models, it does not follow that transfer from more pre-trained models will always improve performance. Whether transfer from multiple pre-trained models improves performance depends on whether the pre-trained models provide complementary features, and an ideal method should benefit from more pre-trained models to the extent that they provide complementary features.
> >
> > We have demonstrated that AFT achieves exactly this: as shown in Figure 2(d) and Figure 3(c), unlike other competing methods, AFT performance directly reflects the performance of the pre-trained features on the downstream task, measured by the linear probe accuracy, for both transfer from single and multiple models. For example, combining features from DINO + CLIP or BiT + DINO + CLIP both achieves higher linear probe accuracy than individual models, and AFT indeed achieves better performance by transfer from the two combinations than from individual models. Conversely, AFT does not improve performance by combining pre-trained models if their features are not complementary and do not improve linear probe accuracy in the first place, as we find in the NLP experiments (Figure 3(c)). We note that our findings agree with the results in [1], which also finds that transfer from more pre-trained models does not always improve performance and recommends using only a small number of pre-trained models in practice.
> >
> > **On results before normalization.** We've added a table of the unnormalized results to Appendix A.4.

---

> > > ### Comment · Reviewer_U9CF · 2023-11-23
> > > **Response to Authors**
> > >
> > > I thank the authors for the response, especially the efforts to provide two more vision datasets and six more language ones. I have some remaining concerns after the rebuttal:
> > >
> > > (1) This paper does not appropriately discuss and compare with existing works on related topics, such as transfer learning and knowledge distillation, which makes its contributions or improvements not convincing enough. Although one additional B-Tuning baseline is added, the compared baselines are far from comprehensive. Only standard fine-tuning and vanilla KD are compared, with more sophisticated transfer learning or KD methods ignored.
> > >
> > > (2) I cautiously disagree with the comments that ‘KD aims to transfer all information in the pre-trained (teacher) features’ since the vanilla KD compared in this paper cannot stand for all KD methods, and the general framework of feature KD mentioned in the previous review already has the potential of selecting relevant features by transforming the features of the teacher model. I agree that ‘not all the pre-trained features will be relevant to the downstream task,’ but simply raising this straightforward idea without a novel technical design cannot be considered a significant contribution.
> > >
> > > (3) Currently, the performance of AFT is still worse than linear probes when the pre-trained models are powerful, and I think this hinders the practical value of this method. Besides, although the pre-trained models in this paper are larger than the downstream ones, they are not too large to run the inference.
> > >
> > > (4) Results of transfer learning from more pre-trained models do not show clear improvements in AFT.

---

### Official Review · Reviewer_U7ws · 2023-11-01

**Soundness:** 3 good
**Presentation:** 3 good
**Contribution:** 2 fair
**Rating:** 6
**Confidence:** 4

**Summary:**

The paper proposes Adaptive Feature Transfer (AFT), a downstream adaptation technique that operates directly on features, thereby decoupling the choice of the pre-trained model architecture from the downstream one. AFT enables combining different pre-trained architectures together during adaptation while distilling only the relevant information for the downstream task to the final model. The algorithm is validated across a diverse set of vision, language and vision-language tasks and compared against knowledge distillation and transfer learning algorithms.

**Strengths:**

1. The proposed method allows to distill features learned with different architectures on possibly different modalities to any given architecture
2. The method is validated on both vision, language and vision-language tasks

**Weaknesses:**

1. The proposed method promises to distill features from **any** set of models to a given model once the downstream task is know. The paper is positioned as a generic method that could be applied to any set of models (possibly containing architectures different to the downstream one). However, while the presented theory to justify the method is sound and generic, the empirical results do not seem to support the claim. For example, in Figure 1 (right) and Figure 2 (b) adding convolutional features to a ViT based downstream model seem to reduce the performance of the model. Why is it the case? To me it seems to suggest that the proposed method is not strong enough to reject some features that will lead to a worse downstream model.
    - If this is the case the current algorithm should be coupled with model selection techniques to pick the best features that are more likely to help (see [1] and reference therein). Can the authors comment on this more?
2. The previous limitation gets even worse when the set of conditioning models gets larger since the signal to noise ratio drops, making extracting the relevant information for the downstream task even harder. I suggest the authors to consider comparing with explicit sparsity inducing methods as the ones proposed in [2] and the references therein.
3. The final algorithm is optimizing theta and rho jointly. However, one would expect \rho being optimized more often than \theta. Typically, this is done with bi-level optimization techniques or simple rewriting \rho in closed form for each given \theta. Did the authors try those more natural alternatives? If \rho is not optimized fast enough the most likely trajectory induced by SGD will be around a stationary point of \rho which leads to a maximally insensitive/uninformative \rho which will be reasonably good on average for many possible \theta, however not optimal for any in particular.


References:

[1] A. Deshpande, et al. “A linearized framework and a new benchmark for model selection for fine-tuning”

[2] M. Fumero, et al. “Leveraging sparse and shared feature activations for disentangled representation learning”

**Questions:**

1. Why should invariance under orthogonal transformation be of help in the practical optimization optimization objective? Can the authors prove how the optimization landscape will change and get easier to optimize? As of now, this intuitive fact, is left to the ablation studies and only supported by empirical observations.
2. Why not using a different kernel than the linear one? This will make the optimization space much smoother (e.g. by choosing a Gaussian kernel).
3. Visual evaluation on CIFAR100 is quite limited, to increase the impact of the paper on the community I suggest the authors to extend the evaluation to other datasets as the ones used in [1].

Minor:
- Some typos and grammatical errors are present in the paper, please proofread the manuscript.
- Can you report in the paper the level of sparsity of the rho projection map? This could help the reader understanding what happens when irrelevant pre-trained models are added to the mix.
- Make the scatter plots with learn probe accuracy vs test accuracy on the same scale. Is the proposed method worse than directly using a linear classifier on the concatenated features?

---

> ### Author Response · Authors · 2023-11-21
> **Author Response**
>
> Thank you for your thoughtful and constructive feedback. To address your questions, we have run several new experiments to verify and investigate the properties of AFT, including ablation studies on the optimization of $\rho$ and visualization of learned $\rho$. We have also significantly expanded the scope of our evaluation to include 8 more datasets. The new results are included in the updated version of the paper. We hope you can consider raising your score in light of our response.
>
> **On further evaluation.**
> Following your suggestion, we include 2 additional vision datasets and 6 additional language datasets from the GLUE benchmark, bringing the total number of datasets to 15. We also report the averaged results in Figure 2(a) and Figure 3(a), showing that AFT significantly outperforms competing methods on average.
>
> **On rejecting less useful features.** In Figure 2(c) (previously Figure 2(b)), transferring convolutional features from a BiT into a ViT downstream model did in fact improve its performance compared to the ViT only standard transfer learning baseline (STL). It's true that transferring from BiT + DINO + CLIP is slightly worse than DINO + CLIP only. However, AFT is successful at upweighting the more useful features even in this case since transfer from BiT + DINO + CLIP is much better than from BiT alone. Indeed, we verify this in Figure 4(a) by visualizing the learned $\rho,$ which are much smaller for BiT features than for DINO and CLIP features when transferring from all three models. To further test AFT's ability to downweight useless features, we run a new experiment where we append random noise features to DINO features and find that AFT learns to significantly downweights the noise features (Figure 4b), and thereby maintains its performance as we increase the number of noise features (Figure 4c) to up to 2048, whereas KD performance quickly degrades.
>
> **On optimizing $\rho$ more often than $\theta$.** By using the kernel formulation and only learning a diagonal $\rho,$ we have considerably reduced the difficulty of optimizing $\rho.$ Instead of optimizing a dense $d_\psi \times d_\psi$ matrix, we only optimize a $d_\psi$ dimensional vector. Furthermore, the bound in Eq 2 is valid even if $\rho$ is not optimal for a given $\theta.$ Therefore, to maximize the number of updates to $\theta$ given a fixed compute budget, we optimize them jointly, exactly analogous to how the encoder and decoder are jointly optimized in VAEs. However, following your suggestion, we added an ablation experiment where we update $\rho$ 5 times per $\theta$ update, and find that it does not improve performance even with the number of updates to $\theta$ fixed (Figure 5(a)).
>
> **On the benefit of invariance under orthogonal transformation.** Invariance of the kernel under orthogonal transformation of the features is helpful because it reduces the number of parameters to learn. By only learning a diagonal $\rho$ with $d_\psi$  parameters, we can effectively search for all transformations $\rho' = U\rho$ for any orthogonal matrix $U$ without having to learn $d_\psi^2$ parameters.
>
> **On using a different kernel.** We have added an ablation experiment where we use a Gaussian (RBF) kernel, and find that it achieves similar but slightly worse performance (Figure 5(a)).
>
> **On AFT performance vs linear probe performance.** We plotted the linear probe accuracy vs test accuracy on different scales for the convenience of visualization because the former has a much larger range than the latter. It’s true that for sufficiently large and strong pre-trained models, linear probe can be better than AFT (and all other methods). However, the linear probe is expensive to deploy, as it requires inference with the original pre-trained model, the very problem that AFT aims to solve, and is therefore not a viable alternative. For example, the linear probe accuracy of ViT-L CLIP roughly matches AFT accuracy when transferred to ViT-S on CIFAR-100. But ViT-L CLIP has 428M parameters, 20 times larger than ViT-S. We have added this discussion to the paper.
>
> **On combining with model/feature selection techniques.** We agree that combining AFT with model selection techniques is a promising direction, and have included a discussion and provided references to the works you mentioned. However, not explicitly performing model selection is not a weakness of AFT. Model selection and AFT are entirely complementary, as AFT solves the problem of how to transfer from any pre-trained model(s), while model selection techniques solve the problem of how to select the best pre-trained models to perform transfer learning from. Similarly, AFT can be combined with feature selection techniques, but not doing so is not necessarily a weakness of AFT. First, transferring only a sparse subset of features is a strong inductive bias that may not be appropriate for all tasks. Second, AFT already learns to downweight the less useful features, as shown in Figure 4.

---

### Official Review · Reviewer_QqHJ · 2023-11-01

**Soundness:** 3 good
**Presentation:** 2 fair
**Contribution:** 2 fair
**Rating:** 6
**Confidence:** 4

**Summary:**

The paper proposes Adaptive Feature Transfer (AFT) to extract information from the (multiple) pre-trained model to the downstream model by minimizing the mutual information between pre-trained and downstream features. The paper at the end uses a stronger regularized loss by only minimizing the feature distance in the downstream and pre-trained space to make the training more robust. The results show that AFT outperforms KD on vision and language tasks and architectures.

**Strengths:**

1. The paper observes that the stronger regularization (using kernels) on the regularization term can further improve the results.

2. The proposed approach outperforms KD on various tasks and architectures.

**Weaknesses:**

1. I do not fully understand what is the main difference between AFT with $\rho$ and KD, namely, the equation (7) and (8). Is the main difference that in equation (7) you downsample the pre-trained features and in equation (8) you upsample the downstream features? If yes, is there mathematical proof (or visualization, other experiments, etc) that this difference really makes the model learn the essential information of downstream tasks and discard useless information?

2. Some parts of Section 3.2 are unclear.

(1) There is a missing $\prime$ in the first kernel definition, the definition of applying the kernel function to vector is undefined in equation (9), $X$ and $X^{\prime}$ should be the same according to Algorithm 1 but not mentioned in the text.

(2) Why the $\rho$ in Section 3.2 does not downsample the feature to the shape of the downstream features ($d_{\phi}$)?

(3) How to optimize U to make sure it is orthogonal?

(4) In Algorithm 1, the definition of $\hat{L}(\theta)$ is missing and $\hat{Y}_{batch}$ is not used.

3. The evaluation is conducted only on small subsets of benchmarks. Using more datasets and reporting the average results would make the results more convincing (like datasets used in few-shot experiments in CLIP, GLUE, SuperGLUE, Winogrande, etc).

**Questions:**

Why choose Eq (7) as the starting point to develop AFT rather than equation (8), as in Figure 4, the results of AFT w/o kernel (optimizing Eq 7 only) are not better than STL (maybe KD either).

---

> ### Author Response · Authors · 2023-11-21
> **Author Response**
>
> We appreciate your constructive feedback. To address your questions, we have significantly expanded our evaluation to both verify the claimed properties of AFT and evaluate its performance on a wider range of datasets. The new results are included in the updated version of the paper. We hope you can consider raising your score in light of our response.
>
> **On further evaluation.** Following your suggestion, we now include 2 additional vision datasets used in CLIP evaluation (DTD and Food-101) and 6 additional language datasets from the GLUE benchmark, bringing the total number of datasets to 15. We also report the averaged results in Figure 2(a) and Figure 3(a), showing that AFT significantly outperforms competing methods on average.
>
> **On the difference between AFT and KD.**
> Conceptually, the main difference between AFT and KD is that KD aims to transfer all pre-trained features to the downstream model, while AFT aims to transfer only the features relevant to the downstream task and is therefore more suitable for transfer learning. As shown in Ahn et al. (2019), the KD objective minimizes a bound on $I(\Psi; X|\Phi)$ (equivalently $H(\Psi|\Phi)$) where $\Psi$ is the pre-trained feature, $\Phi$ is the downstream feature, and $X$ is the input. This objective penalizes any information in the pre-trained features that is not in the downstream features. By contrast, AFT optimizes a bound on $I(\Phi; X|\Psi),$ which only penalizes new information in the downstream features that is not contained in the pre-trained features. In particular, the latter is zero as long as the downstream features are a subset of the pre-trained features, where the opposite is true for the former. As a result, this conceptual difference manifests in the final objectives as whether we train the pre-trained features to be predictable from the downstream features (KD) or train the downstream features to be predictable from the pre-trained features (AFT), as you correctly pointed out.
>
> Following your suggestion, we have added Figure 4 to show that AFT indeed learns to weigh the pre-trained features differently. For example, when we use BiT, DINO, and CLIP as the pre-trained models, AFT upweights CLIP and DINO features and downweights BiT features, which has considerably lower linear probe accuracy (Figure 2d). We also run an experiment where we append random noise features to DINO features and find that AFT learns to significantly downweights the noise features (Figure 4b), and thereby maintains its performance as we increase the number of noise features (Figure 4c), whereas KD performance quickly degrades.
>
> **On choosing the starting point for AFT.**
> As mentioned above, Eq (7) and Eq (8) (now Eq (4) and Eq (5)) are fundamentally different despite their similar forms. The former is conceptually more well-suited for transfer learning and allows selectively transferring the most relevant features. While the ablation study shows that directly optimizing Eq (7) (AFT w/o kernel) is not effective, we show that its kernelized version (AFT) is the best performing method. By contrast, kernelizing Eq (8) is similar to AFT w/o $\rho$ (not allowing weighting the pre-trained features), which we find is less effective.
>
> **On clarifying our definitions.**
> (1) Thank you for catching the typo in the kernel definition. We have fixed it. In Eq 9 (now Eq 6), $k_\Phi(X, X')$ follows the same definition given in the paragraph directly above, namely $\phi(X)\phi(X'),$ where $X$ and $X'$ each denote a single input.
>
> (2) Under the kernel formulation, $\rho$ no longer needs to map the pre-trained features to the shape of the downstream features because the kernel enables comparison between features of different shapes. We have added a sentence to clarify this.
>
> (3) We do not need to actually optimize for an orthogonal $U,$ as a result of using the kernel. Two feature maps $\phi(\cdot)$ and $\phi'(\cdot)$ have the exact same kernel even if they differ by an orthogonal transformation $U^*.$ By contrast, we do need to optimize for the right $U^*$ if we measure their $\ell_2$ distance as in Eq 4.
>
> (4) We updated the text to clarify that the mini-batch loss $\hat{L}(\theta)$ is computed as $L(\theta, Y_{\mathrm{batch}}, \hat{Y}_{\mathrm{batch}})$ where $L$ is some given loss function on the downstream task (e.g. cross-entropy loss).
>
>
> [1] Ahn et al. (2019). Variational Information Distillation for Knowledge Transfer

---

> > ### Comment · Reviewer_QqHJ · 2023-11-23
> >
> > Thank the authors for addressing my concerns, and I have increased my score to 6.

---

### Author Response · Authors · 2023-11-21
**General Response to Reviewers and ACs**

We thank all reviewers for their feedback and support. We provide a general response to highlight the timeliness and significance of the work, address the main questions from the reviewers, and summarize the extensive set of new experiments we have conducted to address the reviewers' suggestions. Those experiments have been included in the updated version of the paper. We believe addressing these suggestions has significantly improved the quality of the paper and we hope our response can be taken into account in the final assessment.

### **On the strengths of our work**

Our work contains 3 particularly impactful contributions:
1. We propose a novel objective for transfer learning that accurately reflects the reality that not all the pre-trained features will be relevant to the downstream task. As a result, this objective is fundamentally more well-suited for transfer learning than Knowledge Distillation (KD), which transfers information irrespective of its relevance to the downstream task.
2. We perform an extensive evaluation of AFT with state-of-the-art pre-trained models and various downstream models on 15 datasets across vision, language, and vision-language tasks, showing that AFT significantly outperforms the competing methods across-the-board and benefits considerably more from stronger pre-trained models.
3. As noted by the reviewers, our work addresses an important and timely problem in transfer learning: how to efficiently transfer a variety of pre-trained models, each requiring increasingly large compute budgets to directly fine-tune and perform inference with, into a single smaller downstream model. We believe our work will enable the community to more effectively leverage foundation models that have otherwise been prohibitively expensive to use.

###  **Addressing the reviewers' main questions**

**On more experiments.**
Inspired by reviewer comments, we have significantly expanded the scope of our evaluation to include 8 more datasets commonly used for transfer learning in vision and language domains, bringing the total number of datasets to 15. We also added B-Tuning [1] as a stronger baseline as suggested by the reviewer. We present the results in Figure 2 and Figure 3 in the updated paper, showing that AFT significantly outperforms competing methods on average.

**On whether more pre-trained models should always help.**
While AFT enables transfer from multiple pre-trained models, it does not follow that transfer from more pre-trained models will always improve performance. Whether transfer from multiple pre-trained models improves performance depends on whether the pre-trained models provide complementary features, and an ideal method should benefit from more pre-trained models to the extent that they provide complementary features.

We have demonstrated that AFT achieves exactly this: as shown in Figure 2(d) and Figure 3(c), unlike other competing methods, AFT performance directly reflects the performance of the pre-trained features on the downstream task, measured by the linear probe accuracy, for both transfer from single and multiple models. For example, combining features from DINO + CLIP or BiT + DINO + CLIP both achieves higher linear probe accuracy than individual models, and AFT indeed achieves better performance by transfer from the two combinations than from individual models. Conversely, AFT does not improve performance by combining pre-trained models if their features are not complementary and do not improve linear probe accuracy in the first place, as we find in the NLP experiments (Figure 3(c)). We note that our findings agree with the results in [1], which also finds that transfer from more pre-trained models does not always improve performance and recommends using only a small number of pre-trained models in practice.

**On whether AFT indeed learns to downweight less useful features.**
We provide two pieces of evidence that AFT indeed learns to downweight less useful features as claimed. First, we show in Figure 4(a) that AFT successfully learns to upweight CLIP and DINO features and downweight the less useful BiT features. Second, we run a new experiment where we append random noise features to DINO features and find that AFT indeed learns to significantly downweight the noise features (Figure 4b) and its performance is unaffected by them (Figure 4c).

**On AFT vs linear probe performance.**
Several reviewers noted that the performance achieved by transferring to a smaller model with AFT is sometimes lower than directly training a linear classifier on the pre-trained features. This is not at all a weakness of AFT because directly using a linear model with the pre-trained features is not a viable alternative, as it requires running expensive inference with the original pre-trained model, the very problem that AFT (and KD) aims to solve by transferring to much smaller models.

[1] Ranking and Tuning Pre-trained Models: A New Paradigm of Exploiting Model Hubs. JMLR 2022

---

### Meta-Review · Area_Chair_QUBP · 2023-12-10

**Metareview:**

This paper presents an approach to adaptively transfer the features extracted from multiple pre-trained models to enable data-efficient downstream task learning. Authors performed a diligent rebuttal, which managed to bump the majority of reviewers up to the positive side.

Since this is a "borderline" paper, I carefully read the paper by myself, with the paper's contributions summarized in Alg. 1. The AC held the opinion more coherent to Reviewer U9CF. There are reasonable concerns that shall be considered more seriously by the authors. By simply searching the literature, there are quite a few papers tackling the problem of "transfer from multiple pre-trained models", such as Zoo-tuning and ZooD. These highly closely related methods, while studying almost the identical problem, were omitted from either literature review or empirical comparison. Second, the authors' opinion on existing KD methods or fine-tuning methods are not well supported. For example, in many "Delta Tuning" methods (e.g. B-Tuning), pre-trained features of higher relevance to the downstream will be transferred more, otherwise less, which is a common practice in the community. Using the "selective" mechanism to distinguish this paper from others is not convincing enough. A paper which cannot put its position right by reviewing the literature comprehensively (after rebuttal) is not acceptable. Third, I think the use of kernel alignment is not well justified. Kernels are rarely used in deep learning anymore, because in many cases they cannot bring benefit over the vanilla methods. So this point cannot be counted as a technical contribution. Fourth, since this paper is highly relevant to ensemble learning, there is a large literature on "distilling from an ensemble," which can be seen similar to the first step of this paper. As seen, while many vision and language datasets are compared, the comparison against the related literature is nonetheless insufficient.

Based on these considerations, the paper needs to be further improved.

**Justification For Why Not Higher Score:**

The paper is not strong enough in the context of "transfer from a zoo of models," in that several related methods are bypassed, while the novelty compared to more sophisticated knowledge distillation (e.g. distill from an ensemble) or transfer learning (e.g. adaptively transfer only related knowledge) is not solid enough.

**Justification For Why Not Lower Score:**

N/A

---

### Decision · Program_Chairs · 2024-01-16

Reject